# CUT-AND-PASTE NEURAL RENDERING

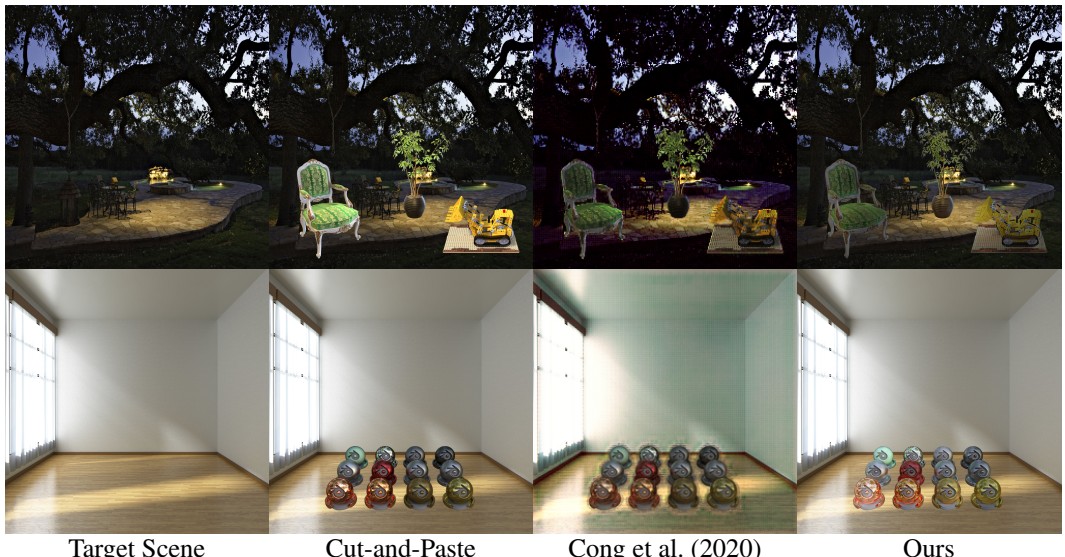

|  Target Scene | Cut-and-Paste | Cong et al. (2020) | Ours |

Figure 1: We generate realistic renderings of cut-and-paste images. Our method is entirely image-based and can convincingly reshade/relight fragments with complex surface properties (a lego dozer, a plant and a chair in top row) and matte, glossy and specular fragments (a set of 16 different materials in bottom row) added to a spatially varying illuminated target scene (indoor-outdoor and day-night) without requiring the geometry of the inserted fragment or the parameters of the target scene.

## ABSTRACT

Cut-and-paste methods take an object from one image and insert it into another. Doing so often results in unrealistic looking images because the inserted object's shading is inconsistent with the target scene's shading. Existing reshading methods require a geometric and physical model of the inserted object, which is then rendered using environment parameters. Accurately constructing such a model only from a single image is beyond the current understanding of computer vision.

We describe an alternative procedure – cut-and-paste neural rendering, to render the inserted fragment's shading field consistent with the target scene. We use a Deep Image Prior (DIP) as a neural renderer trained to render an image with consistent image decomposition inferences. The resulting rendering from DIP should have an albedo consistent with cut-and-paste albedo; it should have a shading field that, outside the inserted fragment, is the same as the target scene's shading field; and cut-and-paste surface normals are consistent with the final rendering's shading field. The result is a simple procedure that produces convincing and realistic shading. Moreover, our procedure does not require rendered images or image decomposition from real images or any form of labeled annotations in the training. In fact, our only use of simulated ground truth is our use of a pre-trained normal estimator. Qualitative results are strong, supported by a user study comparing against state-of-the-art image harmonization baseline.

## 1 INTRODUCTION

Cut-and-Paste rendering involves creating a new image by cutting fragments out of one or more source images and pasting them into a target image; the idea originates with Lalonde et al. (2007). Results are often unrealistic, because of the difference in illumination between the source and target images. But the procedure is useful to artists, and there is consistent evidence that such procedures can be used to train detectors (Liao et al., 2012; Dwibedi et al., 2017). When the geometry and material of the inserted object are known, it is enough to infer an illumination model from the target, render and composite. But current procedures for recovering shape and material from a single fragment simply can't deal with most realistic fragments (think of, say, a furry cat).

This paper describes an alternative method, *Cut-and-Paste Neural Rendering*, that can render convincing composite images by adjusting the cut-and-paste images so that some simple image inferences are consistent with cut-and-paste predictions. So the albedo from the adjusted image should look like cut-and-paste albedo; the shading should look like a shading field; and the image should look like an image. A simple post-processing trick produces very high-resolution composites. Note that all our rendered images are 1024x1024 pixels resolution and are best viewed on screen. Evaluation is mostly qualitative, but we show that our method fools a recent method for detecting tampering.

Our contribution is a method that can realistically correct shading in composite images, without requiring labeled data; our method works for matte, glossy and specular fragments without an explicit geometric or physical model; and human subjects prefer the results of our method over cut-and-paste and image harmonization.

## 2 RELATED WORK

**Object Insertion** starts with Lalonde et al. (2007), who insert fragments into target images. Lalonde et al. (2007) control illumination problems by checking fragments for compatibility with targets; Bansal et al. (2019) do so by matching contexts. Poisson blending (Pérez et al., 2003; Jia et al., 2006) can resolve nasty boundary artifacts, but significant illumination and color mismatches will cause cross-talk between target and fragment, producing ugly results. Karsch et al. (2011) show that computer graphics (CG) objects can be convincingly inserted into inverse rendering models got with a geometric inference or with single image depth reconstruction (Karsch et al., 2014). Inverse rendering trained with rendered images can produce excellent reshading of CG objects (Ramachandran, 1988). However, recovering a renderable model from an image fragment is extremely difficult, particularly if the fragment has an odd surface texture. Liao *et al.* showed that a weak geometric model of the fragment can be sufficient to correct shading *if* one has strong geometric information about the target scene (Liao et al., 2015; 2019). In contrast, our work is entirely image-based: one takes a fragment from one image, drops it into another, and expects a system to correct it.

We use **image harmonization** (IH) methods as a strong baseline. These procedures aim to correct corrupted images. IH methods are trained to correct images where a fragment has been adjusted by some noise process (made brighter; recolored; etc.) to the original image (Sunkavalli et al., 2010; Tsai et al., 2017; Cong et al., 2020), and so could clearly be applied here. But we find those image harmonization methods very often change the albedo of an inserted object, rather than its shading. This is because they rely on ensuring consistency of color representations across the image. For example, in the iHarmony dataset from Cong et al. (2020), they change pink candy to brown (an albedo change; see Fig 12 in Appendix). In contrast, we wish to correct shading alone.

**Image Relighting.** With appropriate training data, for indoor-scenes, one can predict multiple spherical harmonic components of illumination (Garon et al., 2019), or parametric lighting model (Gardner et al., 2019) or even full radiance maps at scene points from images (Song & Funkhouser, 2019; Srinivasan et al., 2020). For outdoor scenes, the sun's position is predicted in panoramas using a learning-based approach (Hold-Geoffroy et al., 2019). One can also construct a volumetric radiance field from multi-view data to synthesize novel views (Mildenhall et al., 2020). However, we do not have access to either training data with lighting parameters/environment maps or multi-view data to construct such a radiance field. Our renderings are entirely image-based. Recent single-image relighting methods relight portrait faces under directional lighting (Sun et al., 2019; Zhou et al., 2019; Nestmeyer et al., 2020). Our method can relight matte, gloss and specular objects with

Figure 2: **Cut-and-Paste Neural Renderer**. Given two images, a target scene (top row) and a new fragment added to this target scene (bottom row), our approach generates a plausible, realistic rendering of the composite image by correcting the fragment's shading. Our method uses DIP as a neural renderer trained to produce consistent image decomposition inferences. The resulting rendering from DIP should have an albedo same as the cut-and-paste albedo; it should have a shading and gloss field that, *outside the inserted fragment*, is the same as the target scene's shading and gloss field. This simple process produces convincing composite rendering for any cut-and-paste images.

complex material properties like cars (Fig 7) for both indoor and outdoor spatially varying illuminated environments only from a single image and without requiring physics-based BRDF (Li et al., 2020).

**Image decomposition**. Land's influential Retinex model assumes effective albedo displays sharp, localized changes (which result in large image gradients), and that shading has small gradients (Land, 1959a;b; 1977; Land & McCann, 1971). These models require no ground truth. An alternative is to use CG rendered images for image decomposition training (Li & Snavely, 2018), particularly with specialized losses (Bi et al., 2015; Fan et al., 2018). One can also train using rendering constraints to produce a form of self-supervised training (Janner et al., 2017). Current image decomposition evaluation uses the weighted human disagreement rate (WHDR) (Bell et al., 2014); current champions are (Fan et al., 2018). We use an image decomposition method built around approximate statistical models of albedo and shading (*paradigms*) to train our image decomposition network without requiring real image ground truth decompositions. Our method has reasonable, but not SOTA, WHDR; but we show that improvements in WHDR do not result in improvements in reshading (Fig 5).

## 3   CUT-AND-PASTE NEURAL RENDERER

We synthesize a reshaded composite image containing a fragment transferred from a source image into a target scene image. We use a deep image prior (DIP) (Ulyanov et al., 2018) as a neural renderer to produce a reshaded image that produces consistent image decomposition inferences. We use an image decomposition trained on *paradigms* (statistical samples of albedo, shading and gloss; Fig 4a) and not real images described in section 3.3, and normals inferred by the method of Nekrasov et al. (2019) to meet the shading consistency tests (section 3.2). The final reshaded image's albedo must be like the cut-and-paste albedo; the reshaded image's shading must match the shading of the target scene outside the fragment; and the shading of the reshaded image must have reasonable spherical harmonic properties and meet a consistency test everywhere Fig 2 summarizes our method.

### 3.1   DEEP IMAGE PRIOR FOR RENDERING CUT-AND-PASTE IMAGES

Assume we have a noisy image $\mathcal{I}_t$, and wish to reconstruct the original. Write $z$ for a random vector, and $f_\theta$ for a CNN with parameters $\theta$ and $E(f_\theta(z); \mathcal{I}_t)$ for a loss comparing the image $f_\theta(z)$ to $\mathcal{I}_t$. The Deep Image Prior seeks

$$\hat{\theta} = \mathrm{argmin}_\theta E(f_\theta(z); \mathcal{I}_t) \qquad (1)$$

and then reports $f_{\hat{\theta}}(z)$. We modify this formulation by requiring that the $E(\cdot; \mathcal{I}_t)$ impose inferential consistency. In particular, write $g_\phi$ for some inference network(s) and $t_\psi(\mathcal{I}_s, \mathcal{I}_t)$ for inferences constructed out of $\mathcal{I}_t$ and the source image $\mathcal{I}_s$. We seek

$$\hat{\theta} = \mathrm{argmin}_\theta E(g_\phi(f_\theta(z)); t_\psi(\mathcal{I}_s, \mathcal{I}_t)). \qquad (2)$$

For us, $g_\phi$ is an image decomposition network (pretrained and fixed), and $t_\psi$ creates target albedos ($\mathcal{A}_t$), shading ($\mathcal{S}_t$) and glosses ($\mathcal{G}_t$) fields. We then train DIP to produce an image that has reasonable intrinsic image properties. For DIP, the input $z$ is the cut-and-paste image and $f_\theta$ is optimized to inpaint inserted fragment and also to meet satisfactory intrinsic image properties.

We use a U-Net with partial convolutions (Liu et al., 2018; Shih et al., 2020; Dundar et al., 2020). However, we find the standard partial convolution often convergence to a trivial solution, producing images close to cut-and-paste and without convincing reshading. To prevent this overfitting to cut-and-paste images, we flip the context for the partial convolution; that is, we consider the inserted fragment as the context and hallucinate/outpaint the entire target scene around it. We can view this as an *inverse partial convolution*.

We use $\mathcal{CP}(\mathcal{I}_s; \mathcal{I}_t; s)$ for an operator that cuts the fragment out of the source image $\mathcal{I}_s$, scales it by $s$, and places it in the relevant location in the target image $\mathcal{I}_t$. $\mathcal{M}$ for a mask with the size of the target image that is 0 inside the fragment and 1 outside. Reconstruction loss for background and is given by:

$$\mathcal{L}_{recons} = \|\mathcal{I}_t \odot \mathcal{M} - (f_\theta(\mathcal{CP}(\mathcal{I}_s; \mathcal{I}_t; s); \mathcal{M}))\|^2 \tag{3}$$

We then pass the DIP rendered image through the image decomposition network $g_\phi$ making $\mathcal{A}_{render}$, $\mathcal{S}_{render}$ and $\mathcal{G}_{render}$ for the albedo, shading and gloss respectively. Our consistent image decomposition inference losses to train DIP are:

$$\mathcal{L}_{decomp} = \|\mathcal{A}_{\mathcal{CP}(\mathcal{I}_s; \mathcal{I}_t; s)} - \mathcal{A}_{render}\|^2 + \|\mathcal{S}_t \odot \mathcal{M} - \mathcal{S}_{render} \odot \mathcal{M}\|^2 + \|\mathcal{G}_t \odot \mathcal{M} - \mathcal{G}_{render} \odot \mathcal{M}\|^2 \tag{4}$$

### 3.2 Shading Consistency Losses

We use two shading consistency losses to make the very strong structure of a shading field apparent to a DIP's rendering. There is good evidence that shading is tied across surface normals (this underlies spherical harmonic models (Liao et al., 2019; Li et al., 2018; Yu & Smith, 2019)), and one should think of a surface normal as a latent variable that explains shading similarities in images. We assume that the resulting illumination, approximated with the first 9 spherical harmonics coefficients ($SHC$; a 9-dimensional vector) does not change when new objects are added to a target scene. We get $SHC$ by solving the least square regression between normals ($\mathcal{N}$) and shading ($\mathcal{S}$) for both the target and the resulting rendered image. We breifly explain the least square regression formulation between normals and shading. Consider we have $k = m \times n$ pixels in an image, our $\mathcal{N} \in \mathbb{R}^{k \times 3}$ and $\mathcal{S} \in \mathbb{R}^{k \times 1}$. We estimate first 9 spherical harmonics basis ($\mathcal{B}(\mathcal{N}) \in \mathbb{R}^{k \times 9}$) from $\mathcal{N}$. We can now write $\mathcal{S} = SHC \times \mathcal{B}(\mathcal{N})$. The solution for $SHC$ is then $\mathcal{B}(\mathcal{N})^\dagger \mathcal{S}$. [1] We get normal estimates from Nekrasov et al. (2019) and then minimize loss ($\mathcal{L}_{SHC}$) between the target and rendered image's SHC($\mathcal{S}; \mathcal{N}$). We use Huber loss for $\mathcal{L}_{SHC}$.

$$\mathcal{L}_{SHC} = \begin{cases} 0.5 \times (SHC(\mathcal{S}_t; \mathcal{N}_t) - SHC(\mathcal{S}_{render}; \mathcal{N}_{\mathcal{CP}}))^2 & \text{for} |SHC_t - SHC_{render}| \leq \mathbf{1}, \\ |SHC(\mathcal{S}_t; \mathcal{N}_t) - SHC(\mathcal{S}_{render}; \mathcal{N}_{\mathcal{CP}})| - 0.5 & \text{otherwise.} \end{cases} \tag{5}$$

Spherical harmonic shading fields have some disadvantages: every point with the same normal must have the same shading value, which results in poor models of (say) indoor shading on walls. To control this effect, we use a neural shading consistency loss ($\mathcal{L}_{NSC}$) that allows the shading field to depart from a spherical harmonic shading field, but only in ways consistent with past inferences. Inspired from a pixel-wise discriminator in Schönfeld et al. (2020), we use a shading consistency network U-Net; $\zeta(\mathcal{S}; \mathcal{N})$, trained to discriminate real and fake shading-normal pair. Our shading-normal consistency network ($\zeta(\mathcal{S}; \mathcal{N})$) produces two outputs; one a pixel level map, yielding the first loss term in Eq. 6 which measures per-pixel consistency; the other an image level value, yielding the second term in Eq. 6 which measures consistency for the whole image. The $\mathcal{L}_{NSC}$ loss is a binary cross-entropy loss. We train DIP using $\mathcal{L}_{NSC}$ to produce a shading field that achieves a pixel-wise consistency score when stacked with the cut-and-paste normals (see Fig 3). Let $m \times n$ be the resolution of our renderings, then $\mathcal{L}_{NSC}$ is given by

$$\mathcal{L}_{NSC} = -\sum_{i=1}^{m}\sum_{j=1}^{n} \log \zeta(\mathcal{S}_{render}[i,j]; \mathcal{N}_{\mathcal{CP}}[i,j]) - log\zeta(\mathcal{S}_{render}\mathcal{N}_{\mathcal{CP}}) \tag{6}$$

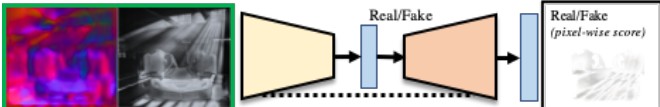

Figure 3: **Shading Consistency U-Net** is trained separately to discriminate consistent and inconsistent pixel-level shading-normal pairs.

In summary, the overall rendering losses that we minimize to update DIP are

$$\mathcal{L}_{render} = \mathcal{L}_{recons} + \mathcal{L}_{decomp} + \mathcal{L}_{SHC} + \mathcal{L}_{NSC} \tag{7}$$

---

[1] $\mathcal{B}(\mathcal{N})^\dagger$ is the pseudo-inverse of $\mathcal{B}(\mathcal{N})$.

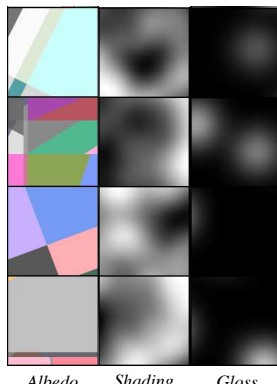 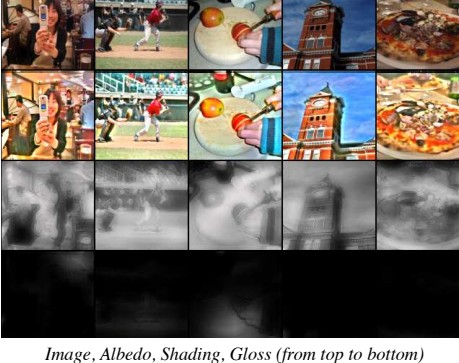

Figure 4: **Image Decomposition.** Left: samples drawn from our *paradigms* that are used to train our image decomposition network. Right: examples showing MS COCO image decompositions after training on paradigms. Note our image decomposition network is trained without real image labeled data.

*Albedo      Shading      Gloss*
*(a) Paradigms*

*Image, Albedo, Shading, Gloss (from top to bottom)*
*(b) Image Decomposition on Real Images (MS COCO)*

### 3.3 INTRINSIC IMAGE DECOMPOSITION

We believe we could use any competent image decomposition network's inferences for reshading. However, our experience suggests accurate albedo recovery (measured by strong WHDR performance) results in poor reshading outcomes (Fig 5). It is also helpful to have a small and efficient network to reduce the overall back-propagation time when training a DIP. Therefore, we trained a small U-Net to produce albedo, shading and gloss layers from images using **paradigms**, samples from statistical models intended to capture the spatial statistics of albedo, gloss and shading. This is a simple extension of the Retinex model (Land, 1977). Albedo paradigms are Mondrian images. Shading paradigms are Perlin noise (Perlin, 1985), and gloss paradigms comprise light bars on a dark background. These are used to produce fake images by a simple composition ($\mathcal{AS} + \mathcal{G}$) that in turn are used to train the image decomposition network. Figure 4a shows some samples from each. As Figure 4b illustrates, the resulting intrinsic image models are satisfactory on MSCOCO real images (Lin et al., 2014). For our experiments, we trained two models (*Paradigm I* and *Paradigm II*); samples drawn from different statistical models, to investigate the consequences of strong WHDR recovery (II has significantly better WHDR than I). The major difference between the two is that Paradigm I has high-frequency, fine-grained details in albedo and not shading, and for Paradigm II it is the opposite.

### 3.4 POST-PROCESSING

**Removing DIP artifacts.** In our approach (see Fig 2), DIP sees both the target image and the cut-and-paste image. We require that DIP inpaints the shading field of the original target image *and* also the cut-and-paste composite image. This process is analogous to rendering scenes once with an object added and once without an object (see (Debevec, 1998; Karsch et al., 2011)). Doing so means the target scene's shading field acts like a regularizer that prevents DIP copying cut-and-paste. Furthermore, we can remove DIP specific noisy artifacts from the reconstructed image. Using the notation above and writing $\mathcal{I}_{obj}$ for the target image rendered by DIP with the object and $\mathcal{I}_{noobj}$ for the target scene rendered by DIP (i.e. no object), we form

$$\mathcal{I}_{final} = (1 - \mathcal{M}) \odot \mathcal{I}_{obj} + \mathcal{M} \odot (\mathcal{I}_t + \mathcal{I}_{obj} - \mathcal{I}_{noobj}) \tag{8}$$

**High-resolution Rendering.** Our image decomposition network can decompose very high-resolution albedos reliably. Since our shading and gloss are both locally smooth for Paradigm I (fine details in albedo), we can easily upsample them to high-resolution. This allows us to render very-high-resolution final reshaded images (1024p) with no additional computational budget. Note that we train DIP to render images with 256p resolution only, and all our final rendered images in this paper are of 1024p resolution. The IH baseline (Cong et al., 2020) can only reshade 256p resolution images.

## 4 EXPERIMENTS

**Dataset.** We collected about 75 diverse images with spatially varying illumination, both indoors-outdoors, day-night to act as target scenes in our experiments. We tested our cut-and-paste rendering on circle cutouts as spheres (Fig 6), plant, chair, lego and a set of sixteen materials used in NeRF

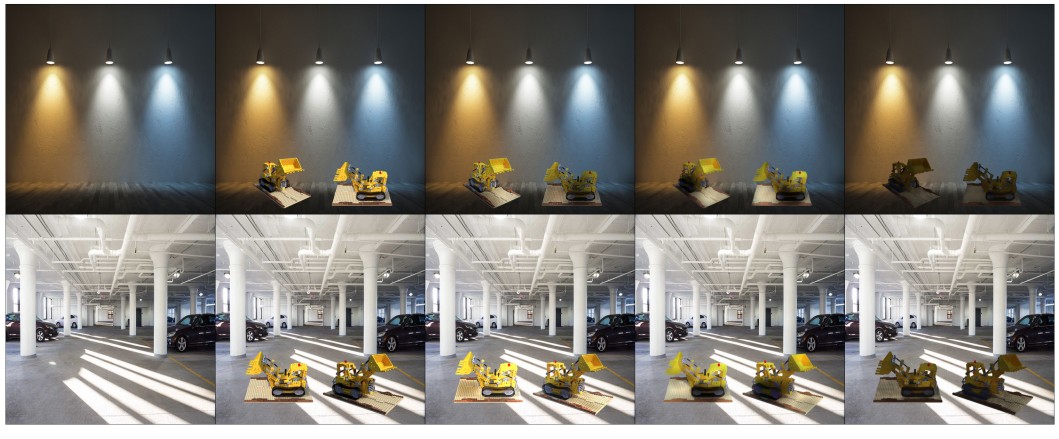

| Target Scene | Cut & Paste | Paradigm I (Our) | Paradigm II (Our) | Li & Snavely (2018) |

Figure 5: Better WHDR does not mean better reshading. Reshaded images when using target inferences from different image decomposition networks. Paradigm I achieves (relatively weak) WHDR of 22%; Paradigm II achieves 19%, close to state-of-the-art (SOTA) for an image decomposition network that does not see rendered images (Liu et al., 2020). However, Paradigm II produces significantly worse reshading results. Furthermore, reshading using CGIntrinsics (Li & Snavely, 2018) (a supervised SOTA method) is also qualitatively worse than using Paradigm I. This reflects that better recovery of albedo, as measured by WHDR, does not produce better reshading. The key issue is that methods that get low WHDR do so by suppressing small spatial details in the albedo field (for example, the surface detail on the lego), but the normal inference method cannot recover these details, and so they do not appear in the reshaded image. From the perspective of reshading, it is better to model them as fine detail in albedo than in shading.

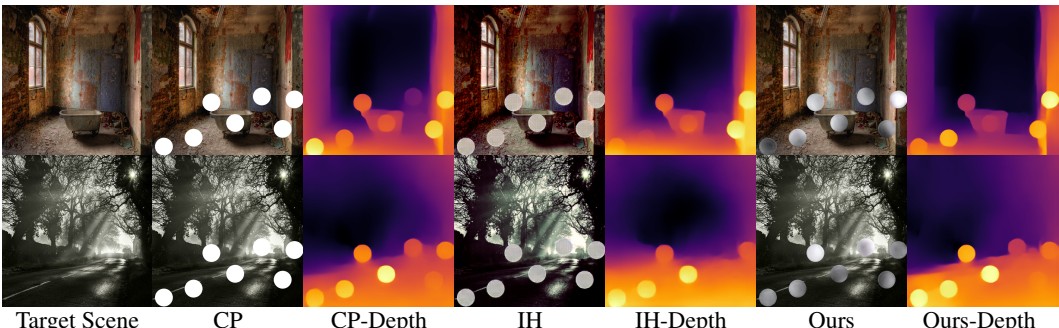

| Target Scene | CP | CP-Depth | IH | IH-Depth | Ours | Ours-Depth |

Figure 6: Our method has some implicit notion of the 3D layout of the scene, which is required to choose the appropriate shading. Our method shades the white discs as spheres (rather better than IH, implying it "knows" about shape). Furthermore, a depth network Ranftl et al. (2020) applied to our reshaded results recovers depths that span the volume of the scene rather more than for others. Note the curious fact that the depth network can choose depths for CP and IH images confidently, too.

(Mildenhall et al., 2020) (Fig. 8). We also show reshading results for cars (Fig 7). Many other objects can be found in our Appendix. We use ADE20k validation set Zhou et al. (2017) for supplying real residual loss to our image decomposition network and also to train our shading consistency network. Note that the ADE20k does not have ground truth surface normals and we use normals from a pretrained network (Nekrasov et al., 2019) as described in section 3.2.

**Training Details.** We used U-Net for DIP, Image Decomposition and Shading Consistency Network. Network architecture and other training details are in our supplementary. We update our DIP for a fixed 10k iterations and this takes about 1200 seconds using our image decomposition network and 1600 seconds when using CGIntrinsic (Li & Snavely, 2018).

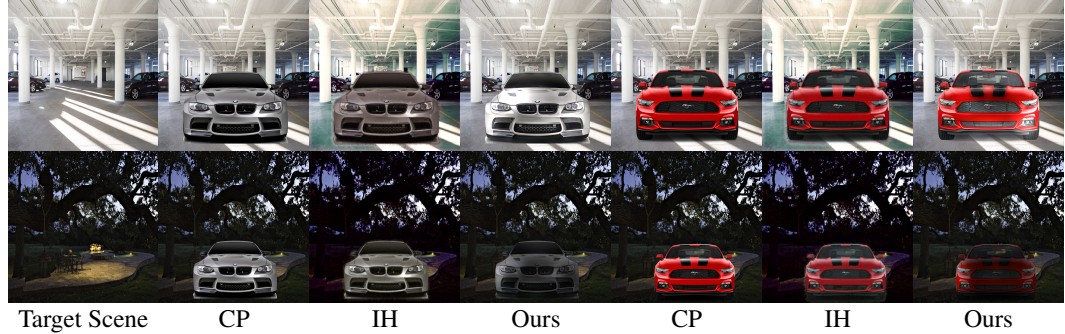

Figure 7: Glossy effects in car paint (with glitter in them) make reshading cars a particularly challenging case with their complex reflective properties. Our method successfully reshades cars without a distinct shift in object and background color produced by IH.

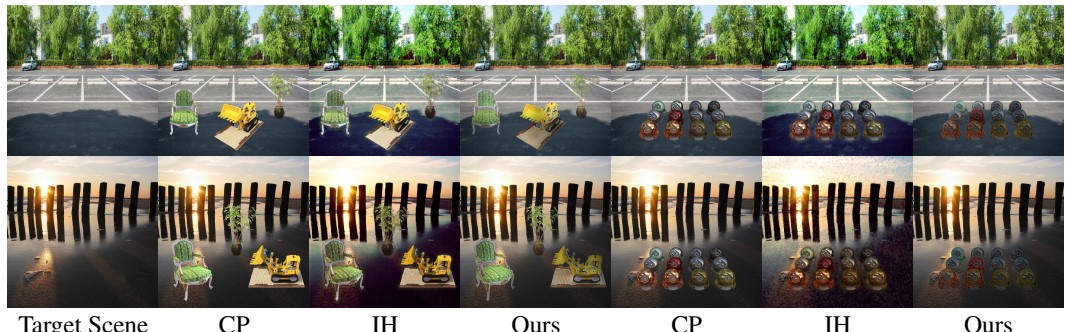

Figure 8: Our method reshades objects with complex surface geometry and complex material properties, and appears to preserve the identity of the material without a distinct shift in object and background color produced by IH.

**IH Baseline.** We use DoveNet from Cong et al. (2020) as our IH baseline. We use their provided pretrained model for our evaluation [2].

**Intrinsic images.** Figure 5 explains how better albedo recovery can lead to weaker reshading. Paradigm I achieves realistic reshading and we use image decomposition model trained on Paradigm I for all our experiments.

**Fooling users.** We wish to know whether our reshaded images are reliably preferred to cut and paste images. We have conducted a user study to compare our reshading method (RS), cut-and-paste (CP), and image harmonization (IH). We collected data from a total of 122 unique users in 500 studies from Amazon Mechanical Turk. Each study consists of a prequalifying process, followed by 9 pair-wise comparisons, where the user is asked which of two images are more realistic. The prequalifying process presents the user with five tests; each consists of an image with inserted white spheres which are not reshaded (i.e. bright white disks) and an image with inserted spheres which have been reshaded (see Fig 6). We ignore any study where the user does not correctly identify all five reshaded images, on the grounds that the difference is very obvious and the user must not have been paying attention. The result is 109 prequalified studies. The comparisons are balanced (each study is 3 RS-IH pairs, 3 CP-IH pairs and 3 RS-CP pairs, in random order and presentation).

The simplest analysis strongly supports the idea that RS is preferred over both alternatives. One compares the probability that RS is preferred to IH (.673, over 327 comparisons, so standard error is .026, and the difference from 0.5 is clearly significant); RS is preferred to CP (.645, over 327 comparisons, so the standard error is .026, and the difference from 0.5 is clearly significant); IH is preferred to CP (.511, over 327 comparisons, so standard error is .027, and there is no significant difference from 0.5). An alternative is a Bradley-Terry model (Tsai et al., 2017; Cong et al., 2020)

---

[2]https://github.com/bcmi/Image_Harmonization_Datasets/tree/master/DoveNet

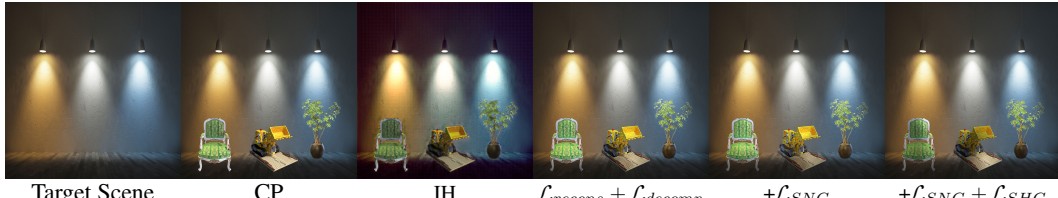

| Target Scene | CP | IH | $\mathcal{L}_{recons} + \mathcal{L}_{decomp}$ | $+\mathcal{L}_{SNC}$ | $+\mathcal{L}_{SNC} + \mathcal{L}_{SHC}$ |

Figure 9: **Ablation study** showing how different components of our loss helps in correcting overall shading. Results improve moving from left to right. We observe that our $\mathcal{L}_{SNC}$ helps in improving local changes based on the near-by surroundings and $\mathcal{L}_{SHC}$ takes into account the overall illumination and hence provides complementary solutions.

used in image harmonization evaluation, regressing the quality predicted by the Bradley-Terry model against the class of algorithm. This yields coefficients of 0 for IH, -0.347 for CP and 0.039 for RS, implying again that RS is preferred over IH and strongly preferred over CP.

**Quantitative Analysis.** We cannot quantitatively evaluate our reshading method, because we do not know the ground truth. But we tried scoring images based on a quality when no reference is available (Mittal et al., 2012). Quite surprisingly, NIQE gets a significantly worse score for target scenes without any fragments added. This shows that getting a reliable quantitative evaluation for reshading is extremely hard. We also tried the recent CNN detector of image fakes described by Wang et al. (2020). We find that it does not detect any of our synthesized images as fake; but it also fails to detect cut-and-paste images and image harmonization images.

| Images | NIQE↓ | CNN-D↓ | B-T↑ |
|---|---|---|---|
| IH | 38.39 | 0 | 0 |
| CP | **9.67** | 0 | -0.35 |
| Ours | 9.78 | 0 | **0.04** |
| Target Scene | 12.51 | 0 | NA |

Table 1: Quantitative Analysis without ground truth is unreliable for reshading. We report numbers for a no-reference image quality estimator (NIQE), a CNN-based tampering detector (CNN-D) and Bradley-Terry (BT) model's analysis from the user-study. Surprisingly, NIQE is worse for target scenes, CNN-D cannot detect any image (CP, IH or ours) as fake, and from the user study between three methods, BT analysis shows our reshaded images are preferred over CP and IH.

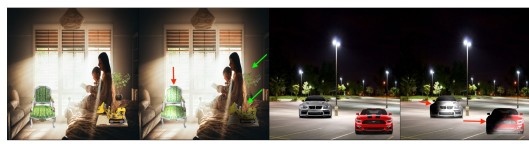

| Cut & Paste | Our | Cut & Paste | Our |

Figure 10: **Failure Examples.** The wrongly shaded regions are marked in red. In the first example, DIP aggressively copies background shading onto the chair. However, the lego and the plant are reshaded accurately. In the second, the reason for failure is surface normals. The scene has two dominant normal directions – ground (pointing upwards) and the sky (pointed towards viewer). The lack of third direction forces DIP to copy shading either from the sky or the ground plane.

## 5 DISCUSSION

**Evaluation on synthetic scenes.** One could build ground truth data on both synthetic (easy, but possibly misleading) and real (experimentally hard, see Karsch et al. (2011)) scenes. But evaluating using these will be misleading. The problem is that it is not possible to recover a renderable representation from an image fragment using current computer vision technology. This means the inserted object will be rendered incorrectly, and so evaluating using a metric like Mean Squared Error (MSE) may be bad. Worse, MSE says nothing about the realism of the reshaded image. One could have good MSE, but poor qualitative results; one could also have bad MSE, but strong qualitative results. Only user study remains as an evaluation.

**Why could this work?** It should be clear that corrections to inserted fragments are not veridical. As Liao et al. (2015; 2019) noted, corrected fragments often fool humans more effectively than physically accurate lighting, likely because humans attend to complex materials much more than to consistent illumination (Berzhanskaya et al., 2005). The alternative physics theory by Cavanagh & Alvarez (2005) argues that the brain employs a set of rules that are convenient, but not strictly physical, when interpreting a scene from retina image. When these rules are violated, a perception

alarm is fired, or recognition is negatively affected (Gauthier et al., 1998). Otherwise, the scene "looks right". This means that humans may tolerate a considerable degree of estimation error, as long as it is of the right kind. By insisting that the image produces consistent inferences, we appear to be forcing errors to be "of the right kind".

**Need For Speed.** A desirable cut-and-paste reshading application would correct shading on inserted objects requiring no explicit 3D reasoning about either fragment or scene. It would do so in ways that produce consistent inferences of image analysis. Finally, it would be quick. We described a method that meets two of these three desiderata (DIP still requires minutes to render an image).

**Casting Shadows.** Our renderer can adjust the background with cast-shadows but often unrealistically. Therefore, we only show shading corrections on the fragment in this paper. Having better inferences (for eg., normal prediction network) should improve our results significantly. Also, better illumination modeling like Lighthouse (Srinivasan et al., 2020) and residual appearance networks (Sengupta et al., 2019) should also improve background shading changes. We also believe GAN based losses should help in correcting background shadows. This can also be combined with a few recent promising work in this direction (Izadinia & Seitz, 2020; Zhan et al., 2020).

**Extending to Videos.** Our method can reshade multiple frames at a time quite convincingly (in Fig 14 of Appendix). Extending to videos is a natural next step. Once the speed of our renderings is improved, we can also use our techniques for improving frame interpolation.

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
