# OpenReview forum: "Cut-and-Paste Neural Rendering"
_ICLR.cc/2021/Conference — Reject_

### Official Review · AnonReviewer3 · 2020-10-16
**Cut-and-paste Neural Rendering**

**Rating:** 5
**Confidence:** 4

**Review:**

This paper proposes cut-and-paste neural rendering that allows to insert objects into a target scene in a plausible manner, i.e., in terms of shading plausibility. At the core of the approach is a deep image prior that allows to match the shading and albedo fields based on shading and albedo consistency losses. A normal estimation network that is trained based on synthetic data is used to further inform shading estimation. The approach is interesting and shows plausible results.

The submission performs many ablation studies that give a better understanding of the inner workings of the approach. The approach is compared to several other methods that are outperformed visually and in a user study. The results obtained by this approach look plausible.

There is one statement in the paper I disagree with: “We cannot quantitatively evaluate our reshading method, because we do not know ground truth”. It is possible in two ways: Completely synthetic scenes that are rendered using computer graphics or real scene where images are taken before and after inserting an object. This would allow access to ground truth pairs of images with and without the object. It would be interesting to see how close the outputs of these methods are to the actual ground truth, both qualitatively and quantitatively. For all shown results, I was always unsure how it would look in reality, which makes it quite hard to judge how good the method actually performs. For sure this approach produces more plausible results than previous approaches, but how close are these results actually to how it would look in reality. I think this question is unanswered so far.

In summary, I like the submission and I think the proposed approach is novel, but I would have hoped for an actual evaluation against GT data. Therefore, I am currently a bit on the fence to suggest acceptance, but would like to hear the opinion of the other reviewers and see the author response.

---

> ### Author Response · Authors · 2020-11-25
> **Response to AnonReviewer3**
>
> We are glad that the reviewer is happy with our novelty and our overall submission. We thank the reviewer for appreciating our efforts on thorough ablations for a better understanding of the inner workings of our method. Also, we are delighted that the reviewer is pleased by our plausible results and also agrees they outperform qualitatively (also supported by our user study). We now address concerns about quantitative evaluation in two parts.
>
> **Evaluation on synthetic scenes:** We agree that one could build ground truth data on both synthetic (easy, but possibly misleading) and real (experimentally hard, see Karsch et. al., 2011) scenes. But quantitative evaluation using these will be misleading.  The problem is that it is not possible to recover a renderable representation from an image fragment using current computer vision technology.  This means the inserted object will be rendered incorrectly, and so the Mean Squared Error (MSE)  may be bad.  Worse, MSE says nothing about the realism of the reshaded image (see notes on human tolerance of shading inaccuracy, below). One could have good MSE, but poor qualitative results; one could also have bad MSE, but strong qualitative results. Only user study remains as an evaluation.  We have conducted a user study on a wide variety of scenes with diverse illumination, objects with complex geometry and shapes, and with as many users as possible to get unbiased human preference evaluations.  We have also updated our paper and added a paragraph on why evaluating on synthetic scenes is misleading in our Sec 5.
>
> **Human tolerance of shading inaccuracy:** We briefly discussed this in our Sec. 5 under "Why could this work?”. The primary goal for any evaluation on this task is to fool humans because the human visual system can tolerate shading inaccuracy. Generating result exactly the same as ground truth isn't the objective, till humans find it 'real'. We go over several studies highlighting that humans are fooled by variations in the 'ground truth'. Humans find it hard to spot inconsistent shadow directions in a single image (Ostrovsky et al., 2005) if gross shading is correct.   Highlights are important material cues for humans (Beck & Slava, 1981), but observers are not perturbed if the highlight is somewhat in the wrong place (see Berzhanskaya et al., 2005, experiment 3). The *alternative physics* theory (Cavanagh & Alvarez, 2005) argues that the brain employs a set of rules that are convenient, but not strictly physical when interpreting a scene from a retina image. When these rules are violated, a perception alarm is fired, or they negatively affect recognition (Tarr et al., 1998). Otherwise, the scene "looks right".  This means that humans may tolerate a considerable degree of estimation error, as long as it is of the right kind.  Our results -- which are clearly neither canonical nor physical, but are effective at fooling human viewers -- support this notion.    We note a poorly understood asymmetry between scene and object here is caused by the way in which methods are used.  Typically, the object that will be inserted is a crucial part of the target image and will command much visual attention from the observer.  This means that humans may react differently to errors in the depiction of objects and scenes.  We are not aware of guidelines in the literature about what is tolerable here.  It is usual to build less scrupulous scene models than object models, and doing so seems not to present difficulties (eg Karsch et. al., 2011; 2014).
>
> References:
> - Yuri Ostrovsky, Patrick Cavanagh and Pawan Sinha. "Perceiving illumination inconsistencies in scenes." Perception 34.11 (2005): 1301-1314.
> - Jacob Beck  and Prazdny Slava. "Highlights and the perception of glossiness." Perception & Psychophysics (1981).
> - Julia Berzhanskaya, Gurumurthy Swaminathan, Jacob Beck, and Ennio Mingolla. Remote effects of highlights on gloss perception. Perception, 2005.
> - Patrick Cavanagh and George A Alvarez. Tracking multiple targets with multifocal attention. Trends in cognitive sciences, 9(7):349–354, 2005.
> - Michael J. Tarr, Daniel Kersten and Heinrich H. Bülthoff. "Why the visual recognition system might encode the effects of illumination." Vision research 38.15-16 (1998): 2259-2275.
> - Kevin Karsch, Varsha Hedau, David Forsyth, and Derek Hoiem. Rendering synthetic objects into legacy photographs. ACM Transactions on Graphics (TOG), 2011.
> - Kevin Karsch, Kalyan Sunkavalli, Sunil Hadap, Nathan Carr, Hailin Jin, Rafael Fonte, Michael Sittig, and David Forsyth. Automatic scene inference for 3d object compositing. ACM Transactions on Graphics (TOG), 2014.

---

### Official Review · AnonReviewer4 · 2020-10-28

**Rating:** 6
**Confidence:** 2

**Review:**

Summary
This paper introduces a reshading method for cut-and-paste image composition. It uses a modified deep image prior as the rendering networking, and trains with a novel shading consistency loss. The results are plausible. Inserted fragments have proper shading with the source image and unchanged albedo.

Pros:
- an improved DIP that produces consistent image decomposition inferences (albedo, shading, gloss).
- a novel shading consistency loss that accounts for the spatial similarity and past inferences.
- Extensive experiments showing advantages over CP and IH.

Cons:
- Is scene geometry not needed for this task?

Cut-and-paste methods may bring ambiguity for the inserted object. Cars in Fig 7 have a fixed depth because it is grounded in human perception. However, the circle cut-outs in Fig 11 are floating, which makes these disks have ambiguous depth. For example, in the second last row of Fig 11, a floating disk closer to the camera will have darker shading while a disk with further depth (under the lamp) will be lit up and has brighter lighting. In this paper, the only supervision on scene geometry is the surface normal in the shading loss. The inserted fragment may have an ambiguous depth in human perception. Therefore, in the inference phase, the shading factor for the cut-out might be ambiguous.

- Quantitative evaluation on albedo

One way to evaluate the reshading method is to measure how much the inserted fragment's albedo changed. Results from CP, RS and IH could be passed into the image decomposition network, and the albedo values could be compared with the ones in the source image.

- Timing for inference

How much time does it take for DIP to render a 1024p image? The paper mentions minutes, but could be more precise.

Comments:
I don't quite follow the spherical harmonics coefficients and how they are calculated from normals and shading. There could be a formulation equation in Sec 3.2 or in supplementary.

Updates: Thanks for the author's response. My concerns are mostly addressed. But I still believe explicit geometry modeling should be included for this task. This could be added in future works. Overall, I am positive on the submission and keep my original rating.

---

> ### Author Response · Authors · 2020-11-25
> **Response to AnonReviewer4**
>
> We are glad that the reviewer liked our extension to deep image prior and help it reason about image decomposition. We also thank the reviewer for acknowledging our shading consistency loss contribution. We expect more follow-ups from the community to further improve our proposed shading consistency losses.  Also, thanks for appreciating our extensive experimental efforts. Now we respond to each concern raised by the reviewer.
>
> **Is scene geometry not needed for this task?**  While we have no explicit representation of scene geometry, there is good evidence that our network has a strong implicit representation. For example, the network renders the spheres of Fig. 6 and Fig. 11 (in supplementary) as well as what it could achieve with added depth information. Moreover, we report a depth-reconstruction network can recover the depth quite accurately (see Fig. 6). However, we currently have no method to control the placement of disks in depth -- the user chooses the 2D location of the disc on the image, and the DIP then assigns shading (and so depth).  The result is consistent and sensible -- there is a plausible depth in the scene such that the rendered sphere could be at this depth and have the shading that it displays. There must be an implicit geometric/depth representation lurking within the deep image prior that helps it to choose correct shading for the inserted fragment. We show disk cutouts to highlight exactly this point that the method appears to understand complex scene geometry, and that’s the reason we can get a plausibly realistic estimate of our reshading.
>
> **Quantitative evaluation on albedo:** Thanks for suggesting us this evaluation. Following are the L2 distance between the rendered albedo and the cut-and-paste albedo from our method and the IH baseline. We will add this to our supplementary.
>
> | IH (Cong et al., 2020) |  Ours |
> |:----------------------:|:-----:|
> |          0.053         | **0.013** |
>
>
> **On timing for inference:** We mention this in our paper. Please see Sec. 4 “Training Details”. We update DIP for a fixed 10000 steps. It requires us about 17 to 20 minutes, depending on the GPU type. The code is not optimized at this point, but we expect it to be around 10-15 minutes.  The main cost is to update the DIP. Once DIP is updated, the final 1024p inference takes 0.03 seconds.
> Note that our DIP is trained to render only 256p resolution images. Only during the final inference, we render high resolution. We also discuss this in Sec. 3 under “Post-processing”.
>
> **On spherical harmonics coefficients formulation:**  We have updated spherical harmonics least square formulation in Sec. 3.2.
>
> References:
> - Wenyan Cong, Jianfu Zhang, Li Niu, Liu Liu, Zhixin Ling, Weiyuan Li, Liqing Zhang. DoveNet: Deep Image Harmonization via Domain Verification. Proceedings of the IEEE/CVF Conference on Computer Vision and Pattern Recognition (CVPR), 2020, pp. 8394-8403

---

### Official Review · AnonReviewer2 · 2020-10-29
**Carefully designed pipeline with well conducted experiments**

**Rating:** 6
**Confidence:** 4

**Review:**

**Paper Summary**
The paper proposed a carefully designed neural rendering pipeline to realistically insert one image fragment into another image via deep image prior and the consistency in the albedo, shading, and gloss in the rendered image. Experiments show great improvement over baselines both qualitatively and quantitatively (via user study).

**Strength**
1. The inverse partial convolution is a neat idea. This idea effectively helps the network to learn the shading of the target image and to avoid directly copying the inserted objects.
2. The shading consistency loss and the designed experiments to pre-train the image decomposition network are also carefully designed and thoughtful. The analysis in Fig 5 also provides insights for the relationship between WHDR and the actual rerendering effects. This could inspire many followups on how we should design the algorithm to better composite an image.
3. The experiments are also well performed. To quantitatively evaluate the performance, this paper conducted user studies with detailed descriptions of the whole procedure. And qualitatively, the results the paper showed are better than baselines.

**Weakness**
1. Speed, the overall pipeline relies on the training Deep Image Prior (DIP), making the whole pipeline slow during the real application.
2. Some notation in this paper is not clear. E.g.  in Eq. 6,  what does the second term means?
3. The paper didn't provide the exact method for Image Harmonization baseline, it would be better for the readers to understand what's the exact method to be compared.
4. It's not clear how does the proposed method generalize to high-order lighting effect since the lighting model the paper used is Spherical Harmonic.

**Overall**
Overall, I think this paper proposed many neat ideas for image composition and performed solid experiments, but considering the limitation of the method (which has been described in Sec. 5), I vote for a weak accept initially.

---

> ### Author Response · Authors · 2020-11-25
> **Response to AnonReviewer2**
>
> We are glad that the reviewer likes our inverse partial convolution and our image decomposition idea. We are also happy to see the reviewer found our experiments to be comprehensive. We now answer the reviewer’s concerns one by one:
>
> **1. Speed:** We acknowledge in our paper (see Sec. 5) that our current method is limited in rendering time by the use of Deep Image Prior (DIP). However, we have a follow up in progress to make our pipeline real-time. Our first work in this direction is a proof of concept; it is possible to reason about complex images intrinsic only from a single 2D image and get a realistic reshading outcome. In our next ongoing follow up, our primary focus is rendering reshaded images in real-time without losing perceptual quality.
>
> **2. Eq 6:**  Our shading-normal consistency network ($\zeta$ in Eq. 6) produces two outputs: (a) a pixel level map, yielding the first term, which measures per-pixel consistency, and (b) an image-level value, yielding the second term, which measures consistency for the whole image. We have updated the paper for more clarity.
>
> **3. Image harmonization:** We accept this feedback and have updated the paper (in Sec. 4) to cite the paper in Sec. 4, besides the previous reference in Fig. 1. We use the recent CVPR 2020 state-of-the-art method, DoveNet (Cong et al., 2020) as our Image Harmonization baseline. DoveNet was trained on several color corrupted datasets using a supervised method; translating color corrupted fragments to harmonized/non-corrupted fragments. We used their provided pretrained model for our evaluations.
>
> **4. Higher-order lighting effects:**  The rendering literature seems to agree with us that spherical harmonics is still a very good approximation for modeling lighting effects (see Liao et al., 2019; Li et al., 2018; Yu & Smith, 2019; Zhou et al., 2019; Nestmeyer et al., 2020).  One disadvantage of spherical harmonic lighting is flat surfaces have constant shading, which is unrealistic; but we have a shading consistency loss (Eq. 6) which allows deviations from spherical harmonic shading as long as they are overall consistent with the network’s experience of shading.  Adversarial losses on the rendered images might improve shading, contact shadows, and cast shadows further.
>
> References:
> - Wenyan Cong, Jianfu Zhang, Li Niu, Liu Liu, Zhixin Ling, Weiyuan Li, Liqing Zhang. DoveNet: Deep Image Harmonization via Domain Verification. Proceedings of the IEEE/CVF Conference on Computer Vision and Pattern Recognition (CVPR), 2020, pp. 8394-8403
> - Zicheng Liao, Kevin Karsch, Hongyi Zhang, and David Forsyth. An approximate shading model with detail decomposition for object relighting. International Journal of Computer Vision, 2019.
> - Zhengqin Li, Zexiang Xu, Ravi Ramamoorthi, Kalyan Sunkavalli, and Manmohan Chandraker. Learning to reconstruct shape and spatially-varying reflectance from a single image. ACM Transactions on Graphics (TOG), 2018.
> - Ye Yu and William AP Smith. Inverserendernet: Learning single image inverse rendering. In Proceedings of the IEEE Conference on Computer Vision and Pattern Recognition, 2019.
> - Hao Zhou, Sunil Hadap, Kalyan Sunkavalli, and David W Jacobs. Deep single-image portrait relighting. In Proceedings of the IEEE International Conference on Computer Vision, 2019.
> - Thomas Nestmeyer, Jean-Francois Lalonde, Iain Matthews, Andreas Lehrmann. Learning Physics-guided Face Relighting under Directional Light. Proceedings of the IEEE/CVF Conference on Computer Vision and Pattern Recognition (CVPR), 2020, pp. 5124-5133

---

### Decision · Program_Chairs · 2021-01-07
**Final Decision**

**Decision:**

Reject

**Comment:**

Overall the review is borderline: R2 and R4 are slightly positive and R3 is slightly negative. All the reviewers like the novel shading consistency loss proposed in the paper and, improved DIP that produces consistent image decomposition inferences, and good experimental results. However, reviewers also shared concerns about speed and the thoroughness of the evaluation, and human tolerance of shading inaccuracy. These points were addressed in details in the rebuttal, and reviewers didn’t change their initial scores.

The AC is concerned about the cut-and-paste neural rendering results. Because there are no cast shadows, the rendering doesn’t look realistic under the lighting conditions in the new image. It’s unclear that the proposed method would lead to a promising direction of copying and pasting contents into images for photorealistic editing. Consequentially, the paper is not ready for publication at its current form.